# Assessment of Inter-Sectoral Virtual Water Reallocation and Linkages in the Northern Tianshan Mountains, China

**Dedao Gao [1,†], Aihua Long [2,*,†], Jiawen Yu [3,†], Helian Xu [1,*], Shoujuan Su [4] and Xu Zhao [5]** 

[1]  School of Economics & Trade, Hunan University, Changsha 410000, China; gdd1995345@163.com
[2]  State Key Laboratory of Simulation and Regulation of Water Cycle in River Basin, Department of Water Resources, China Institute of Water Resources and Hydropower Research, Beijing 100038, China
[3]  College of Water and Architectural Engineering, Shihezi University, Shihezi 832000, China; yujiawen_415@163.com
[4]  College of Geography and Environment Science, Northwest Normal University, Lanzhou 730000, China; 18394170228@163.com
[5]  Institute of Blue and Green Development, Shandong University, Weihai 264209, China; xuzhao@sdu.edu.cn
*   Correspondence: ahlong@iwhr.com (A.L.); xuhelian@163.com (H.X.)
†   These authors contributed equally to this study and share first authorship.

**Abstract:** Quantitative analysis of the reallocation and linkages of virtual water in the economic sector was important for the integrated water resources management in inland arid regions. Taking the northern Tianshan Mountains (NTM) as an example, we applied the environmental input-output model to design the accounting framework for the reallocation of blue and green virtual water (VW) in the economic sector and analyzed the correlation effect of VW reallocation among various sectors by backward and forward linkages in economic analysis. The results showed that the direct blue and green water consumption of primary industry respectively accounted for 99.2% and 100% of the total water consumption in NTM. Planting sector had the largest amount of VW outflow among all sectors. Animal husbandry, forestry and construction had a large pulling effect on VW outflow of planting sector, while planting sector and animal husbandry were the main sectors for VW export of blue and green water. We suggest that the government can increase the import of blue-green VW for agricultural raw materials through VW trade and develop industries such as service and electricity that have less pulling effect on the primary industry VW, so as to improve the economic added value of VW in the primary industry and reduce the loss of VW in primary industry production and trade flows in future water management.

**Keywords:** blue water; green water; water shortage; normalized backward linkages; internal allocation

## 1. Introduction

Water shortage is an important issue restricting the sustainable development of the global economy and society. In the context of globalization, competition between industries and uneven economic development have exacerbated the contradiction between water supply and demand in some water-scarce countries or regions [1]. It is generally believed that the increasing human demand for goods and services is the main driver of water shortages [2]. Traditional water management has mainly focused on the physical water (PW) of economic sectors, such as agriculture, manufacturing and mining industries. In the last two decades, the academic research on "virtual water" has provided a new way of thinking for improving the level of water management in water-scarce areas and alleviating water shortages. Virtual water (VW) is water embodied in the production process for both intermediate use

and final demand [3]. PW and VW are interrelated. When PW is consumed in the production or service, it turns into VW, and the VW flow is reallocated among various economic sectors through supply chains [4]. Import of VW through trade can help to alleviate local PW shortages [5]. For example, the Chinese government included VW into the national water safety strategy in order to guarantee the national water security and seek the coordinated development of PW and VW in 2017 [6]. At present, water resources that can be directly used by humans and ecosystems mainly include blue water (surface and ground water) and green water (water stored in unsaturated soil layers and canopy evapotranspiration from rainfall) [7]. Green water is an important water resource for maintaining global food production and ecosystem service functions. According to statistics, 83% of the food production depends on rain-fed agriculture dominated by green water globally, and in southern Africa this proportion was even more than 90% [8]. In addition, some researchers believe that improving the use efficiency of green VW for production is an effective measure to alleviate the pressure of blue VW resources in water-scarce regions [9,10].

The existing VW accounting researches have been conducted from two perspectives. One focused on the VW flow of products or services among different regions and analyzed the relationship between socio-economic activities and water consumption at global, national and regional scales [11], with most of the study areas located in water-scarce countries or regions [12]. Another perspective focused on inter-sectoral VW flows and linkages for one single region, and the meaning of these studies is to support decision making in balancing the trade-offs between regional economic development and water resource depletion. For instance, Zhao et al. (2016) found that a production structure adjustment would increase the "internal water use of products," mainly due to a shift from agricultural and industrial sectors to service sectors in Beijing, China [13]. Zhang et al. (2019) believed that the economic structure should be adjusted from increasing agricultural production to increasing the economic value of the embodied VW and reducing the export of the VW from low economic value products [14]. Yu et al. (2010) found that the forward reallocation linkage of the service industry in the southeast and northeast of the UK was greater than the agriculture sector, indicating that the development of the local services needs to be obtained from goods and services of other water-intensive sectors, which has led to an increased in the total water consumption of the UK [15]. The above studies show the importance of analyzing VW reallocation and linkages among sectors to pinpoint the key sectors that pull the limited PW input [16]. However, less studies analyze the VW reallocations and linkages among different economic sectors for both blue and green water. Especially for arid regions, the increasingly fierce competition for blue and green water resources between agriculture and other industrial sectors has become a major constraint on economic growth in arid and semi-arid regions [17].

In this study, we applied the combination of environmental input-output analysis and the CROPWAT model to propose a VW reallocation accounting framework for typical inland arid regions dominated by an agricultural-leading economy, and the CROPWAT model is a computer program for the calculation of crop water requirements and irrigation requirements based on soil, climate and crop data. The study area is the northern Tianshan Mountains, China (NTM, Figure 1), which consists of Changji, Bozhou, Ili, Tacheng, Altay, Urumqi, Karamay, Turpan and Hami regions. Our research here distinguishes from the previous studies by calculating the direct consumption of blue PW and green PW in various economic sectors and analyzing reallocation impact after conversion into blue VW and green VW. Normalized forward reallocation linkages, normalized backward reallocation linkages and internal reallocation can clearly reflect the virtual water correlation effect between industries in the study area.

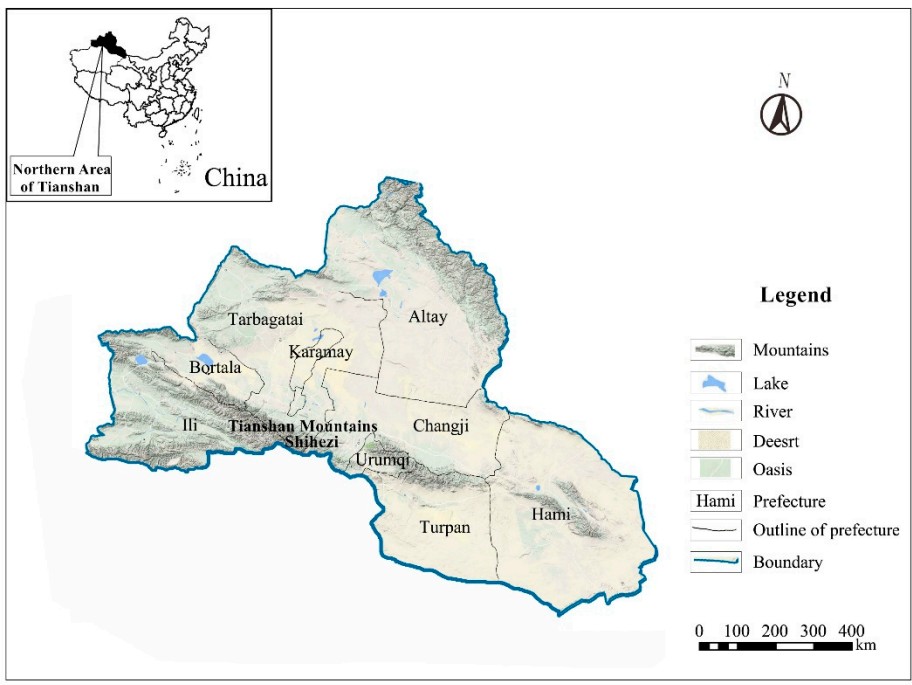

**Figure 1.** Geographical location map of the northern Tianshan Mountains (NTM) region.

## 2. Methodology and Data

### 2.1. Study Area

The NTM region is located in the Eurasian continent, with a total area of $60.26 \times 10^4$ km$^2$. It has a typical temperate continental arid climate, characterized by low rainfall and extremely limited water resources. The annual average evaporation is 1500–2700 mm, and the average annual rainfall is only 150–200 mm. In 2016, the total precipitation was 121.03 km$^3$ and the total annual renewable water resources were 44.43 km$^3$. The total population was 12.41 million, and the annual total GDP was 90.04 billion USD, of which the tertiary industry had the highest share (43.40 billion USD). The value added of the secondary and primary industry was 37.94 billion USD and 8.69 billion USD, respectively. The NTM borders Mongolia, Russia, Kazakhstan, and Kyrgyzstan.

### 2.2. Methodological and Empirical Framework

#### 2.2.1. Update and Compilation Method of Input-Output Table

The British economist Richard Stone proposed the concept of the bi-proportional scaling technique in 1961, namely the RAS method. The RAS method is a biproportional technique, which adjusts the original matrix by rows and columns with uniform multipliers [18]. The RAS method can revise the direct consumption coefficient matrix in the original input-output table through data such as total output, initial input, and total final demand, and compile the input-output table during the planning period based on the matrix. We used the RAS method to update the 2012 input-output table of Xinjiang compiled by the Xinjiang Statistics Bureau to year 2016. The RAS method has the advantages of low data cost and fast table compilation speed [19]. It has been widely used in the update and compilation of input-output tables in many countries and regions [20,21]. The specific update steps of the input-output table in the NTM are shown in Figure 2. Meanwhile, according to the classification standard of sectors in the input-output table [22], the 42 sectors classified in the input-output table are combined into nine sectors to match with the sectoral water data (Appendix A, Table A1).

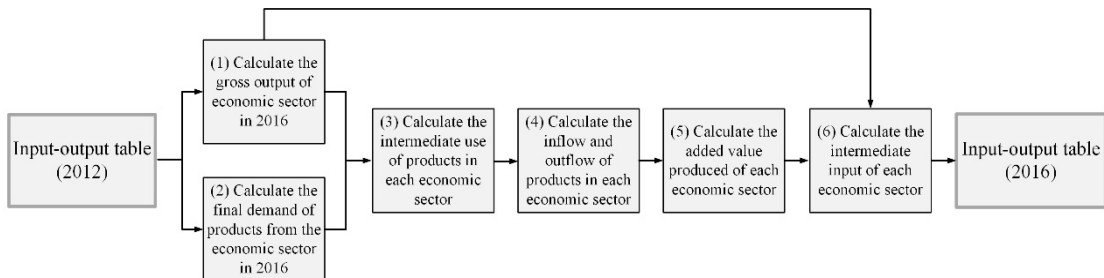

**Figure 2.** The updated path of the input-output table in the NTM region based on the RAS method.

2.2.2. Virtual Water Reallocation Accounting Methods

The basic equations of the environmental input-output model are shown below, following [23]:

$$X = AX + Y \tag{1}$$

where $X$ is the sectoral output vector, $A$ is the technical coefficient matrix, which reflects the direct consumption of unit products produced by a certain sector to the products of other relevant economic sectors, $Y$ is the final demand vector which includes the internal consumption vector of region and export vector. Based on the above method, we can obtain the virtual water allocation matrix among economic sectors. First, we calculate the reallocation matrix among sectors after blue PW and green PW are converted to VW. The equation is shown as follows:

$$V = D \cdot L \cdot Y = \frac{b + g}{x} \cdot (I - A)^{-1} \cdot (f + e) \tag{2}$$

where $V$ is the allocation matrix among economic sectors after PW are converted to VW, $b$ and $g$ are blue and green PW consumption vector of products directly produced by each sector, $x$ is the sectoral output vector, $D$ is the direct water consumption intensity matrix, $I$ is the unit matrix, $L = (I - A)^{-1}$ is the Leontief inverse matrix, $(f + e)$ is the final demand vector, $f$ is the internal consumption vector of region, including rural consumption, urban consumption, government consumption and total capital formation, $e$ is the export vector (including export to domestic and export to foreign countries).

For $n$ sectors, Equation (1) can be expressed in matrix form as follows:

$$\begin{bmatrix} v^{11} & v^{12} & \cdots & v^{1n} \\ v^{21} & v^{22} & \cdots & v^{2n} \\ \vdots & \vdots & \ddots & \vdots \\ v^{n1} & v^{n2} & \cdots & v^{nn} \end{bmatrix} = \begin{bmatrix} d^1 & 0 & \cdots & 0 \\ 0 & d^2 & \cdots & 0 \\ \vdots & \vdots & \ddots & \vdots \\ 0 & 0 & \cdots & d^n \end{bmatrix} \begin{bmatrix} l^{11} & l^{12} & \cdots & l^{1n} \\ l^{21} & l^{22} & \cdots & l^{2n} \\ \vdots & \vdots & \ddots & \vdots \\ l^{n1} & l^{n2} & \cdots & l^{nn} \end{bmatrix} \begin{bmatrix} y^1 & 0 & \cdots & 0 \\ 0 & y^2 & \cdots & 0 \\ \vdots & \vdots & \ddots & \vdots \\ 0 & 0 & \cdots & y^n \end{bmatrix} \tag{3}$$

For the allocation matrix $V$, the sum $v^i$ of all elements in any row $i$ represents the total amount of VW allocated by local blue and green water to the $i^{th}$ sector, and the diagonal element $v^{ii}$ represents the internal allocation of VW by sector $i$, any element $v^{ij}$ $(i \neq j)$ in row $i$ represents the VW transferred from sector $i$ to sector $j$. Therefore, by removing the diagonal elements in the allocation matrix, the inter-sector VW reallocation matrix $T$ can be obtained as:

$$T = \begin{bmatrix} 0 & v^{12} & \cdots & v^{1n} \\ v^{21} & 0 & \cdots & v^{2n} \\ \vdots & \vdots & \ddots & \vdots \\ v^{n1} & v^{n2} & \cdots & 0 \end{bmatrix} \tag{4}$$

where the total amount of VW reallocated from any sector *i* to other sectors can be expressed as:

$$ve^i = \sum_{j=1}^{n} v^{ij}, i \neq j \tag{5}$$

The total amount of VW reallocated from any sector *j* to meet its final demand can be expressed as:

$$vi^j = \sum_{i=1}^{n} v^{ij}, i \neq j \tag{6}$$

In addition, because we use the single-region input-output table and the water consumption coefficient of imported products is difficult to obtain, we assume that imported products and local products have the same water consumption coefficient. This assumption has been applied in many single-region input-output studies [24,25]. Therefore, this paper calculates the VW imports, exports and net imports of different economic sectors:

$$V_N = V_e - V_m = D \cdot L \cdot e - D \cdot L \cdot m \tag{7}$$

where $V_e$ is the sub-sector VW export matrix, $V_m$ is the sub-sector VW import matrix, $V_N$ is the sub-sector VW net import matrix, *m* is the sub-sector import vector.

### 2.2.3. Green Physical Water Consumption in Planting Sector

The CROPWAT model is a decision support tool developed by the Food and Agriculture Organization of the United Nations (FAO). It is widely used to simulate irrigation requirements ($IR_k$) and effective rainfall ($ER_k$) based on meteorological data, crop characteristics and soil physical and chemical properties [26]. At present, this model has been extensively used in many fields such as calculating crop water requirements and developing irrigation schedules [27,28]. We used this model to simulate the $IR_k$ and $ER_k$ of 15 major crops in the NTM, including rice, wheat, coarse cereals, soybeans, cotton, oil plants, sugar beets, vegetables, melons, potatoes, alfalfa, grapes, apples, fragrant pears and red jujube. The share of the cultivated areas of these crops to total planting areas was 91.28% in 2016. We assumed only the planting sector directly consumes green PW for crop production. We are aware that blue and green water consumptions can be interrelated. For example, deep percolation or the existence of irrigation networks fed from groundwater or natural courses (blue water) can change hydrological features, leading to changes in green water consumption [29,30]. In this study, the impact of blue water consumption on the changes of green water consumption is not considered. Therefore, the blue and green PW consumption of the planting sector was estimated according to the sum of $IR_k$ and $ER_k$ of the 15 crops:

$$w_b = \frac{\sum\limits_{k} (IR_k \times h_k)}{\sum\limits_{k} h_k} \times S_T \tag{8}$$

$$w_g = \frac{\sum\limits_{k} (ER_k \times h_k)}{\sum\limits_{k} h_k} \times S_T \tag{9}$$

where $w_b$ and $w_g$ are the blue PW and green PW consumption of the planting sector, $IR_k$, $ER_k$ and $h_k$ are the irrigation requirement, effective rainfall and cultivated area of crop *k*, respectively, and $S_T$ is the total cultivated area of study area.

### 2.2.4. Backward and Forward Virtual Water Linkages

Backward and forward linkages are one of the main methods for analyzing inter-sectoral interconnections in the input-output model [31]. Backward linkages refer to the correlation

between an economic production sector and its upstream sectors (such as providing raw materials, auxiliary materials, energy). Forward linkages refer to the correlation between an economic sector and its downstream sectors (such as the production sector that uses or consumes products from this sector). Through backward and forward linkage analysis, we can identify major economic sectors in a complex industrial linkage [32]. The backward (or forward) linkages can be further divided into internal redistribution and backward (or forward) reallocation linkages. Among them, internal redistribution reflects the input provision and derived demand for other sectors, while the backward (or forward) reallocation linkages reflect a sector's "backward (or forward) dependence" on or linkage to the rest of the economic sectors [23]. The equations are as follows:

$$BL(d)_j = BTL(d)_j + IDL(d)_j \tag{10}$$

$$FL(d)_i = FTL(d)_i + IDL(d)_i \tag{11}$$

where $BL(d)_j$ is backward linkages, $BTL(d)_j = \sum_{i=1}^{n} d^i \times l^{ij} (i \neq j)$ is backward reallocation linkages, $IDL(d)_j = \sum_{i=1}^{n} d^i \times l^{ii}$ is internal backward redistribution. $FL(d)_i$ is forward linkages, $FTL(d)_i = \sum_{j=1}^{n} d^i \times l^{ij} (i \neq j)$ is forward reallocation linkages, $IDL(d)_i = \sum_{j=1}^{n} d^i \times l^{jj}$ is internal forward redistribution.

Combining the concepts of backward and forward linkages, we identified main VW reallocation-in and reallocation-out sectors by analyzing the correlation between industrial VW. Backward, forward and internal reallocation linkages were normalized, respectively. The equations are as follows:

$$\overline{BL}(d)_j = \frac{BL(d)_j}{(1/n)\sum_{j=1}^{n} BL(d)_j} = \frac{\sum_{i=1}^{n} d^i \times l^{ij}}{(1/n)\sum_{i=1}^{n}\sum_{j=1}^{n} d^i \times l^{ij}} \tag{12}$$

$$\overline{FL}(d)_i = \frac{FL(d)_i}{(1/n)\sum_{i=1}^{n} FL(d)_i} = \frac{\sum_{j=1}^{n} d^i \times l^{ij}}{(1/n)\sum_{j=1}^{n}\sum_{i=1}^{n} d^i \times l^{ij}} \tag{13}$$

$$\overline{IDL}(d)_i = \frac{IDL(d)_i}{(1/n)\sum_{i=1}^{n} IDL(d)_i} = \frac{\sum_{j=1}^{n} d^i \times l^{jj}}{(1/n)\sum_{j=1}^{n}\sum_{i=1}^{n} d^i \times l^{jj}} \tag{14}$$

where $\overline{BL}(d)_j$ is normalized backward linkages, $\overline{FL}(d)_i$ is normalized forward linkages, $\overline{IDL}(d)_i$ is normalized internal allocation. $d^i$ is the direct water consumption intensity of sector $i$, $l^{ij}$ is the element of row $i$ and column $j$ in the Leontief inverse matrix.

### 2.3. Data

The data used to update the NTM input-output table in 2016 including the total output, final consumption, gross capital formation and import-export volume, etc. were collected from the Xinjiang Statistical Yearbook (2016) and the statistical yearbooks of counties or regions in Xinjiang (Supplementary Materials Tables S1–S4). In the primary industry, the blue PW consumption data for Forestry, Animal husbandry and Fishery were collected from the Xinjiang Water Resources Bulletin (2017). The blue PW consumption data for the secondary industry and the tertiary industry were calculated based on the "water consumption per unit of GDP," "water consumption per unit of

value-added," "water consumption quota of industry" and "output of major products," etc. in the Xinjiang Statistical Yearbook (2016) (Supplementary Materials Table S5). The crop growth parameters were derived from the CROP database of the FAO (Supplementary Materials Table S6). The CROP database is comprehensively cover production of 173 primary crops for all countries and regions in the world, and data are expressed in terms of area harvested, production quantity and yield. See Appendix A, Table A1 for the sectors included in the primary, secondary and tertiary industries. The effective irrigation area, crop yield and other production data were derived from the Xinjiang Statistical Yearbook (2016). Meteorological data such as precipitation, sunshine hours and temperature were collected from 38 weather stations in NTM.

## 3. Results

### 3.1. Internal Virtual Water Reallocation

The reallocation of blue and green VW between various economic sectors is shown in Figure 3. The results showed that the planting sector had the largest amount of blue water for internal allocation in NTM (4414.1 × 10$^6$ m$^3$), accounting for 53.9% of its blue PW consumption, and the remaining 46.1% of blue water was mainly reallocated to animal husbandry, construction and forestry sectors. Compared with the planting sector, other sectors had much less blue PW consumption. Blue PW consumption of animal husbandry and forestry were 1349.0 × 10$^6$ m$^3$ and 1292.3 × 10$^6$ m$^3$, respectively, and blue water consumed for internal allocation in these two sectors accounted for 76.4% and 99.2% of their total blue PW consumption, respectively. Blue PW consumption of other economic sectors was less than 300 × 10$^6$ m$^3$, of which blue water used for internal allocation in manufacturing, mining and separating and services was less than 50% of total consumption. Especially for services, blue PW consumption was 10.5 × 10$^6$ m$^3$, blue water used for internal reallocation only accounted for 29.7%, and the remaining 70.3% of blue water mainly flowed into other economic sectors with services in the form of virtual water. In addition, the planting sector consumed almost all green PW of NTM, of which 2752.8 × 10$^6$ m$^3$ was used for internal reallocation of sectors, and the amount of green water reallocation to animal husbandry, construction and forestry was higher than the amount of reallocation to other sectors (574.1 × 10$^6$ m$^3$, 507.2 × 10$^6$ m$^3$ and 438.4 × 10$^6$ m$^3$, respectively).

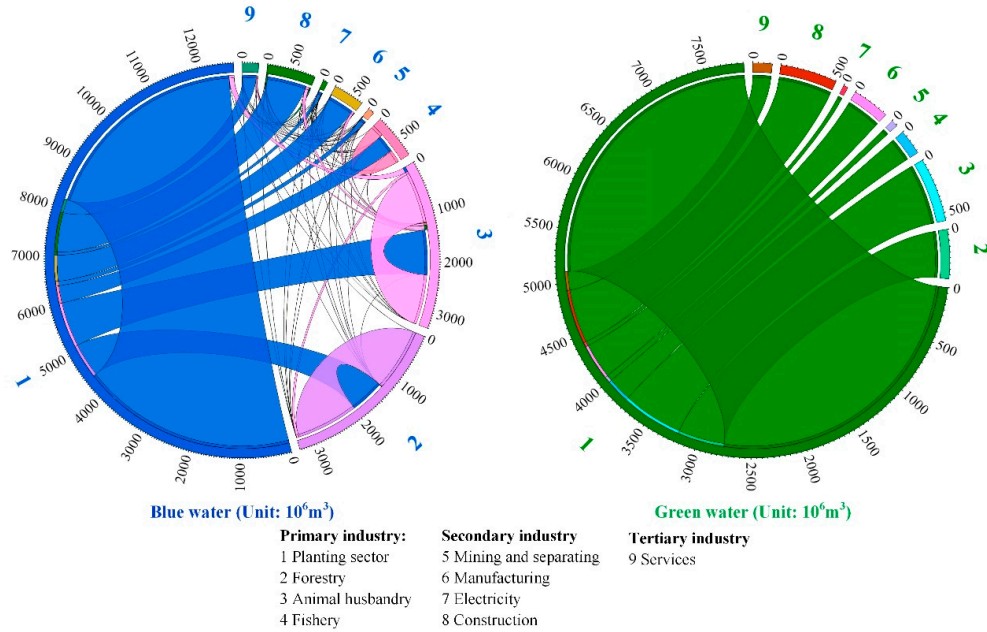

**Figure 3.** Reallocation of blue water and green water between different economic sectors in NTM.

We aggregated the nine economic sectors into three industries. The reallocation of VW among the three industries is shown in Table 1. The results showed that blue PW consumption of NTM was 11.162 km$^3$, of which 17.6% was used for VW reallocation between industries (1.965 km$^3$) and 82.4% was used for intra-industry allocation (9.135 km$^3$). The reallocated PW accounted for 17.5%, 18.1% and 70.0% of PW from primary, secondary, and tertiary industries, respectively. Only 0.011 km$^3$ of blue VW flowed from a secondary and tertiary industry to primary industry to meet its final demand, whilst 1.640 km$^3$ and 0.304 km$^3$ flowed to secondary and tertiary industries, respectively, to meet their final demand. Green PW consumption of NTM was 5.104 km$^3$, of which 21.9% of the primary industry's PW was reallocated to other industries (1.115 km$^3$) and 78.1% was used for intra-industry distribution (3.990 km$^3$). Secondary and tertiary industries received 0.941 km$^3$ and 0.174 km$^3$ of green VW reallocation from the primary industry, respectively.

**Table 1.** Virtual water reallocation matrix among industries in NTM (Unit: km$^3$).

| | Industry | Primary Industry | Secondary Industry | Tertiary Industry | PW Consumption | Internal Distribution |
|---|---|---|---|---|---|---|
| Blue water | Primary industry | 9.135 | 1.640 | 0.304 | 11.079 | 9.135 |
| | Secondary industry | 0.008 | 0.059 | 0.005 | 0.072 | 0.059 |
| | Tertiary industry | 0.003 | 0.004 | 0.003 | 0.010 | 0.003 |
| | VW consumption | 9.146 | 1.703 | 0.313 | 11.162 | 9.197 |
| Green water | Primary industry | 3.990 | 0.941 | 0.174 | 5.104 | 3.990 |
| | Secondary industry | 0.000 | 0.000 | 0.000 | 0.000 | 0.000 |
| | Tertiary industry | 0.000 | 0.000 | 0.000 | 0.000 | 0.000 |
| | VW consumption | 3.990 | 0.941 | 0.174 | 5.104 | 3.990 |

Overall, the total blue PW consumption of the secondary industry was 0.072 km$^3$, but the secondary industry has driven the reallocation of 1.644 km$^3$ of blue VW from the other two industries, accounting for 96.5% of blue VW required for its final demand. The driving effect of the tertiary industry to other industries was similar to the secondary industry. Therefore, the reallocation of VW in NTM has increased the amount of VW required by secondary and tertiary industries to produce final demand products.

*3.2. Backward and Forward Linkages of Virtual Water*

The backward and forward linkages reflect the economic interconnection of various economic sectors due to final demand [33]. According to Equations (12)–(14), the normalized matrix of backward and forward reallocation linkages for water resources in economic sectors can be further divided as blue water and green water (Appendix A, Tables A2 and A3). Figure 4 reveal the normalized backward and forward linkages for blue water and green water. It can be seen from Figure 4 that the economic sectors where the normalized direct backward linkages of blue VW and green VW were both higher than 1.0, including the planting sector (blue water: 2.78, green water: 3.39), forestry (blue water: 2.00, green water: 1.40), animal husbandry (blue water: 1.55, green water: 1.27) and construction (blue water: 1.37, green water: 1.57), which means that the backward linkage strength of these four sectors was higher than the average level of all sectors, and they drove other sectors to reallocate more VW to them.

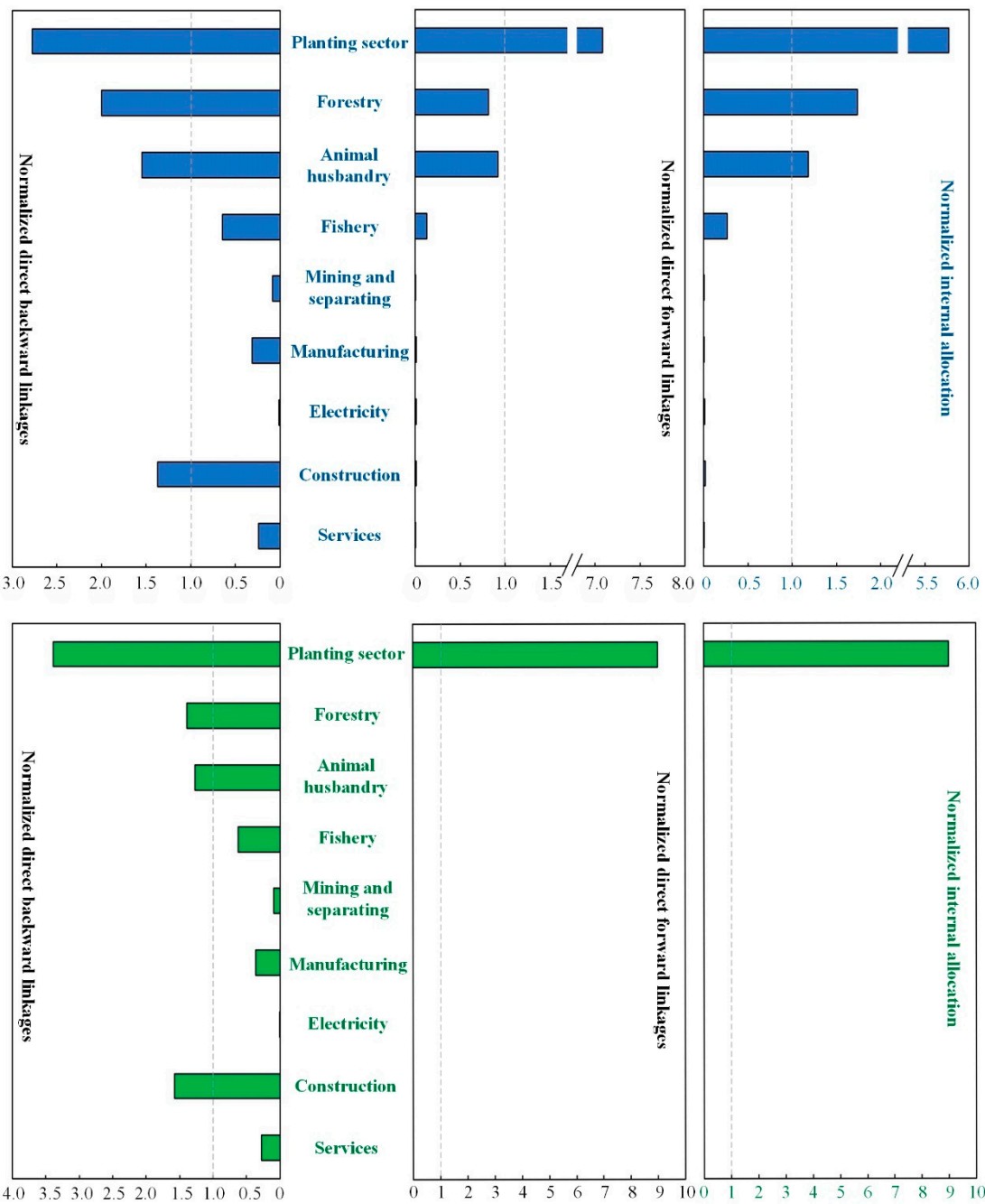

**Figure 4.** Normalized backward and forward linkages and internal allocation of economic sectors.

The forward linkages refer to the promotion effect of an economic sector on the use of VW in all sectors including its own sector, and the decomposed forward linkages can clearly reflect the promotion effect of a sector on the use of VW in other sectors. According to traditional forward linkage equations (the sum of forward reallocation linkages and internal allocation), the planting sector was the absolute dominant sector in forward reallocation linkages, and the normalized direct forward reallocation linkages of blue and green water in the planting sector were 7.08 and 9.00, respectively. Normalized forward reallocation linkages of other sectors were much lower than the planting sector. For example, animal husbandry was the second largest normalized direct forward reallocation linkage of blue water; its value was only 0.92. Green water normalized direct forward reallocation linkage of other sectors was zero. For internal allocation, the larger economic sectors of blue water included

the planting sector, forestry and animal husbandry, indicating that the direct water resource input required by these three sectors in the production process was relatively large (Supplementary Materials Tables S7–S9).

### 3.3. Inter-Industrial Flow, Import and Export of Virtual Water

Figure 5 shows the VW inter-industry flow, import and export in NTM. In 2016, blue VW and green VW in the region were net export status. The net export volume of blue VW and green VW was 3.973 km$^3$ and 2.195 km$^3$, respectively. The import volume of blue VW and green VW was 0.477 km$^3$ and 0.258 km$^3$ from the outside, and export volume of blue VW and green VW was 4.451 km$^3$ and 2.452 km$^3$. Combining the results in Section 3.1, we can infer that the large demand for local blue-green PW in the primary industry sector drives VW export in NTM. Primary industry products establish a strong connection with the other local sector products through blue-green PW consumption, resulting in difficulties on the nonlocal agricultural products flowing into the NTM, and on NTM importing water-intensive agricultural products from other places. Export of a large number of water-intensive agricultural products would further aggravate the scarcity of local water resources so that the pressure of local economic production on water resources would be difficult to export to the water-rich regions outside NTM through trade.

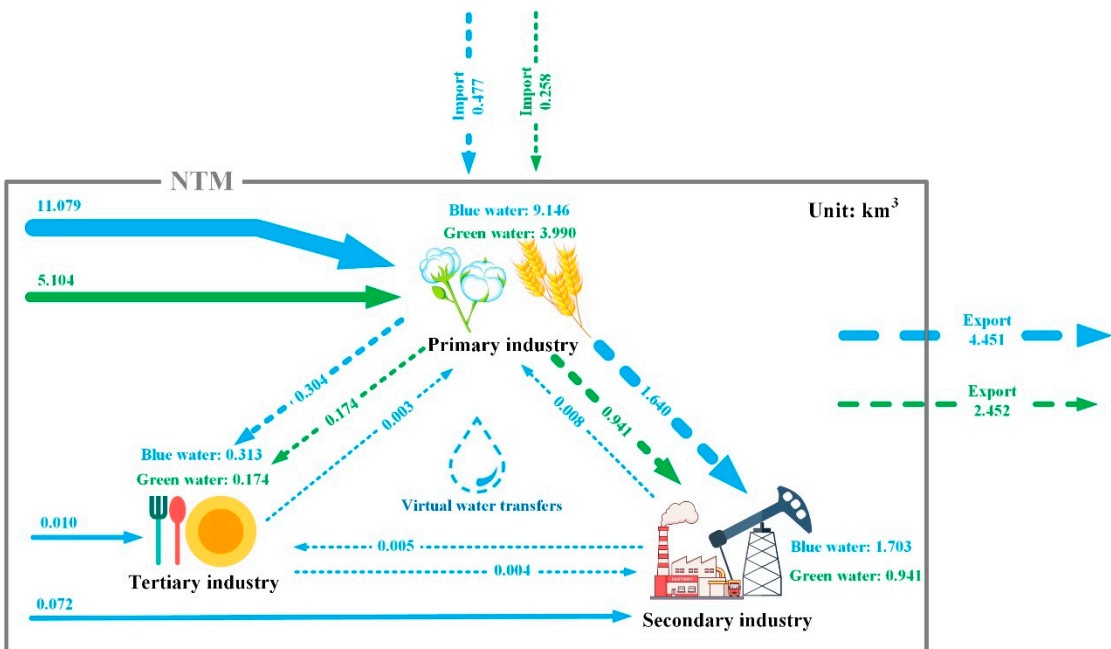

**Figure 5.** Virtual water inter-industry flow, import, and export in NTM.

From the results of VW trade between local economic sectors and external regions (Table 2), the planting sector was the largest blue-green net VW export sector of NTM, and the net VW net export of this sector accounted for 81.6% of the total in NTM. VW net exports of animal husbandry, mining and separating were much higher than other sectors. In addition, manufacturing and construction were net VW import sectors, but the overall net import volume of VW was relatively low (ranging from 0.064 km$^3$ and 0.017 km$^3$). It can be seen that a large part of blue-green water was used for VW export in the planting sector. The above results indicate that export of blue-green VW from the planting sector accounted for 41.0% and 40.1% of local blue-green PW consumption, which means that a large part of blue-green PW consumed by the planting sector flows to the regions outside NTM in the form of VW trade export.

**Table 2.** Net blue-green virtual water export in NTM with sector details (Unit: km$^3$).

| Sectors | Planting Sector | Forestry | Animal Husbandry | Fishery | Mining and Separating | Manufacturing | Electricity | Construction | Services |
|---------|-----------------|----------|------------------|---------|------------------------|----------------|-------------|--------------|----------|
| Blue water | 3.128 | 0.031 | 0.619 | 0.006 | 0.158 | −0.041 | 0.018 | −0.011 | 0.067 |
| Green water | 1.905 | 0.007 | 0.182 | 0.002 | 0.083 | −0.023 | 0.009 | −0.006 | 0.037 |

The reasons for VW trade structure mentioned above include: (i) Climate conditions of NTM were suitable for the cultivation of grain crops and cotton. The average annual grain yield from 1988 to 2015 was $5.79 \times 10^6$ t and the average annual cotton yield was $7.82 \times 10^5$ t, among which cotton export volume accounted for more than 70% of total cotton output [34]. As the NTM region is one of China's major grain and cotton production areas, the local production of high-water-consumption cash crops dominated by cotton consumes a large amount of PW, while the development of local manufacturing and services also relies on these products. (ii) Due to the low level of industrialization in NTM, the local industrial raw materials and products are insufficient and need to be imported from other regions.

## 4. Discussion

Inland arid regions are facing freshwater resources competition due to their increasing human activities [35,36]. The water consumption conflict between the water-intensive economic sector represented by the planting sector and other sectors is particularly prominent. Taking a typical inland arid region in China as an example, we analyzed the reallocation of blue-green VW between various economic sectors and their linkage effects, and then clarified corresponding regulation methods. Our observation has a positive effect in understanding the water utilization status and alleviating the water competition among sectors in the region and has the potential to solve the dilemma of regional economic development with water supply and demand contradiction.

Our results show the necessity to adjust relevant policies according to the current industrial development structure of NTM and its VW reallocation characteristics. According to the results of this study, the direct consumption of blue-green PW in the primary industry accounted for 99.2% and 100% of total water consumption, respectively. The planting sector had the largest amount of VW outflow among all sectors. For the direct PW consumption of the planting sector, 39.7% of blue water was driven by final demand of other economic sectors. It can be seen from normalized backward reallocation linkage results of various sectors that the planting sector had the largest backward reallocation value of blue-green water in all sectors, while construction, forestry and animal husbandry also had a large pulling effect on VW reallocation from the planting sector. Although the direct water consumption of construction was relatively low, the development of this sector was also increasing local agricultural and livestock products output, which in turn increased the indirect water consumption of this sector [37]. In the NTM region, except for construction, the manufacturing and services had a pulling effect on the VW consumption of the local primary industry. We recommend that local government pay attention to the pulling effect of manufacturing and services on primary industry VW consumption in the future and adjust corresponding policy measures to effectively control the adverse effects of this pulling effect.

With regard to the supply of raw materials, we recommend that the government strictly control the choice of local producers of their raw materials in the upstream supply chain and supervise the use of their products in the downstream supply chain. Restrict producers who cause large consumption of local water resources or adversely affect the local ecological environment due to the supply of agricultural raw materials [38]. From the perspective of government, when the study area is developing or attracting external investment in secondary and tertiary industries, it is necessary not only to consider the future economic development potential of these sectors, but also to focus on examining the correlation effects among the local primary industry. They can also assess the amount of VW consumed by these production sectors upstream of their supply chain using local agricultural and livestock

products as raw materials, and the impact on local water shortages. According to impact degree, these foreign capitals are required to pay corresponding benefit compensation to the NTM region (including project investment, capital compensation, trade facilitation and information sharing, etc.), and local government can use the compensation to improve water-saving technology of agricultural products in production and processing process. In addition, the government can use existing industries to establish an integrated environmental protection practice platform in the early stage, combined with the "Water Stewardship" project launched by World Wide Fund for Nature (WWF), so the government can encourage and guide all sectors to strengthen cooperation with various stakeholders in the upstream and downstream industry chain. Local government should improve the water management level of NTM by establishing a "Government-Economic Sector-Public" joint participation model and reduce water security risks faced by various economic sectors and their stakeholders. Economic sectors or enterprises should be willing to participate in the platform when it is piloted, and this water management model should be further promoted locally. In the NTM region, the development of secondary and tertiary industries can drive the increase in demand for agricultural raw materials, thereby increasing PW consumption in the primary industry during local production processes. The NTM region is located in the core area of "The Belt and Road" initiative. With this geographical advantage and policy advantages, it will also be more conducive to inflow agricultural products and raw materials from other regions in the future. For the local primary industry, the government needs to avoid water use problems caused by its excessive development. The government can also increase employment opportunities for farmers through employment training and guidance, so that farmers can also participate in the development of local secondary and tertiary industries.

When using the input-output method for VW accounting, most of the past studies only considered the blue VW reallocation process and paid little attention to the reallocation of green VW between industries and regions [14]. In this study, the CROPWAT model was used to calculate green water consumption in NTM. It was believed that green PW direct consumption was only in the planting sector but green VW had been redistributed among various local sectors in the form of VW reallocation, indicating that green water was also an indirect resource necessary for production in other economic sectors. Green water is a kind of water resource restricted by the region; unlike blue water, it cannot implement cross-regional dispatch of PW through water reallocation engineering measures. At present, the cross-regional redistribution and management of green water can only be achieved through VW trade [39]. Therefore, relevant sectors of NTM should emphasize green VW import contained in VW products. In addition, we recommend that the government develop local industries such as electricity and services that are less driven by VW from the planting sector. Especially in services, with the gradual implementation of the "Tourism Industry Revitalizing Xinjiang" strategy in NTM, it is recommended that the NTM region focus on the development of cultural sports, accommodation and catering, wholesale and retail, and transportation in the future. The direct PW consumption in these sub-sectors is relatively low, and these sub-sectors supplying indirect water to the planting sector is relatively low compared to other secondary and tertiary industry sectors. Hence, taking the services as a main direction of local social and economic development in the future will also help to save local blue-green water in NTM.

Although the present study elucidates the inter-sectoral VW reallocation and linkages in the NTM, some shortcomings due to the limitations on data acquisition remain and certain analysis methods require further research. One of the limitations is that grain crops and economic crops dominate agricultural production in inland arid regions [40]. Therefore, we did not consider the amount of green water directly consumed by the forestry sector. We assumed that only the planting sector directly consumes green PW for crop production, and the impact of blue water consumption on green PW consumption was not considered. The second limitation is due to the application of a single-region input-output method. This paper only provides information on water consumption among economic sectors within NTM but is unable to provide detailed results of water consumption among economic sectors beyond the NTM. The third limitation is that this paper did not distinguish groundwater and

surface water within the blue PW consumption by all sectors. According to the statistics, groundwater consumption by economic production within the NTM increased from 3.976 km$^3$ in 2006 to 6.599 km$^3$ in 2016, indicating that there is a rapidly increasing exploitation of groundwater in the NTM by economic production [41]. Therefore, future studies should distinguish between the dependence of local economic sectors on surface water and groundwater.

## 5. Conclusions

In arid and semi-arid regions, it is crucial for local economic development to accurately identify the driving factors for water use of economic sectors, and understand how much PW in each economic sector is converted to VW and the redistribution process to various sectors driven by final demand. Hence, the framework can be used as an analysis tool to guide water management in inland arid regions. Whether in other inland arid regions in China or in countries or regions with similar conditions, it can provide a more detailed reference basis for local water management reform and water safety construction.

In this study, we revealed the reallocation and linkage effects of VW in the production of various economic sectors of NTM and, based on the combination of environmental input-output analysis and CROPWAT model, we provided a comprehensive framework to account for the reallocation of blue-green water between various economic sectors. The framework can help to understand the reasons for local water pressure and differences in the VW reallocation pulling effect between industries for an agriculture dominated region. Specifically, compared with secondary and tertiary industries, the consumption of blue-green PW and reallocation of blue-green VW in the primary industry constitutes the main body of water resource distribution in the study area. The reallocation of local VW between industries has increased the amount of VW required by secondary and tertiary industries to produce final demand products. Meanwhile, the development of secondary and tertiary industries has also driven the consumption of VW in the local primary industry. Except the planting sector, the direct consumption of blue PW in animal husbandry and forestry was not only higher than other secondary and tertiary industries in NTM, but they also drove the planting sector to reallocate more VW to them. Construction and manufacturing also have a relatively large pulling effect on the VW reallocation from the local planting sector, but they were both in the state of net import of blue-green VW. Electricity and services had little pulling effect on the local planting sector VW. In the process of implementing the VW strategy, we recommended that governments control the development scale of the primary industry and increase the import volume of agricultural product raw materials required by secondary and tertiary industry production processes, and focus on the development of industries with relatively low direct PW consumption and VW pulling effects, such as electricity and services.

In the VW trade flow, NTM is a net export region of blue-green VW. Because the exported VW was mainly contained in the products produced by the planting sector, the amount of reallocation was much higher than other sectors, hence, this VW trade pattern will increase local water pressure in the short term. In the long run, this trade pattern is also not conducive to local social stability and economic development. The proposed analysis framework can be integrated well with local industrial development policies, can help the government identify the water-saving potential sectors in the region, and can provide a quantitative basis for the formulation of future industrial structure adjustment programs. Future research can extend the current framework to a multi-region input-output analysis framework to compare the relationship of VW reallocation between the region and external regions under bilateral or multilateral trade, which promotes a more systematic and comprehensive research perspective than the current analysis framework.

**Supplementary Materials:** The following are available online at http://www.mdpi.com/2073-4441/12/9/2363/s1, Table S1: Total output of each economic sector, Table S2: Total output of planting sector, Table S3: The internal consumption and export of each economic sector, Table S4: The technical coefficient matrix of each economic sector, Table S5: Blue and green physical water consumption of economic sectors, Table S6: Blue and green physical water consumption of planting sector, Table S7: The backward and forward virtual water linkages of each economic sector, Table S8: The normalized backward and forward linkages of each economic sector, Table S9: The normalized internal allocation of each economic sector.

**Author Contributions:** D.G., J.Y. and X.Z. wrote the manuscript text and prepared the figures. A.L. and H.X. advised the study design and data analyses. J.W. and S.S. collected the data and provide shape files of figures. A.L. and X.Z. revised the paper. All authors have read and agreed to the published version of the manuscript.

**Funding:** This research was supported by the National Key Technology R&D Program of China (2017YFC0404300), (2016YFA0601602).

**Acknowledgments:** We are in great debt to Xiaoya Deng and Jie Wang for many helpful discussions and comments on the manuscript. We thank anonymous reviewers and the editor for helpful discussions and comments on previous versions of the manuscript.

**Conflicts of Interest:** The authors declare no conflict of interest.

## Appendix A

**Table A1.** Compilation of economic sectors in the socio-economic system.

| Code | 42 Sectors in the Original IO Table | Aggregated 9 Sectors | Code | 42 Sectors in the Original IO Table | Aggregated 9 Sectors |
|---|---|---|---|---|---|
| 1 | Agriculture | Planting sector | 23 | Electricity, steam and hot water production and supply | Electricity |
| | | Forestry | 24 | Gas production and supply | |
| | | Animal husbandry | 25 | Water production and supply | |
| | | Fishery | 26 | Construction | Construction |
| 2 | Coal mining | Mining and separating | 27 | Transport, storage and post services | Services |
| 3 | Petroleum and natural gas extraction | | 28 | Wholesale and retail trade services, | |
| 4 | Metal ore mining | | 29 | Accommodation and food serving services | |
| 5 | Non-metal mining | Manufacturing | 30 | Information transfer and software engineering | |
| 6 | Manufacture of food products and tobacco processing | | 31 | Information technology service | |
| 7 | Textiles, Wearing apparel, leather, fur, down and related products | | 32 | Finance | |
| 8 | Sawmills and furniture | | 33 | Real estate | |
| 9 | Paper and products minerals, printing and record medium reproduction | | 34 | Leasing and business service | |
| 10 | Petroleum processing, coking and nuclear fuel processing | | 35 | Scientific research and technical services | |
| 11 | Chemical industry | | 36 | Environmental and public facilities management | |
| 12 | Nonmetallic mineral products | | 37 | Resident service | |

**Table A1.** *Cont.*

| Code | 42 Sectors in the Original IO Table | Aggregated 9 Sectors | Code | 42 Sectors in the Original IO Table | Aggregated 9 Sectors |
|---|---|---|---|---|---|
| 13 | Metal smelting, pressing and relatedproducts | | 38 | Education services | |
| 14 | General machinery | | 39 | Health care and community service | |
| 15 | Special purpose machinery | | 40 | Facility repair service | |
| 16 | Transport equipment | | 41 | Public management and social security | |
| 17 | Electric equipment and machinery | | 42 | Culture, sports and entertainment | |
| 18 | Electronic and telecommunication equipment | | | | |
| 19 | Instruments and meters | | | | |
| 20 | Other manufacturing products | | | | |
| 21 | Scrap and waste | | | | |
| 22 | Metal products, machinery and equipment repair services | | | | |

**Table A2.** Normalized matrix of backward and forward reallocation linkages for blue water in economic sectors.

| Sectors | Planting Sector | Forestry | Animal Husbandry | Fishery | Mining and Separating | Manufacturing | Electricity | Construction | Services | Forward Linkages |
|---|---|---|---|---|---|---|---|---|---|---|
| Planting sector | 2.6694 | 1.0995 | 0.9992 | 0.4915 | 0.0755 | 0.2861 | 0.0050 | 1.2390 | 0.2177 | 7.0830 |
| Forestry | 0.0022 | 0.8041 | 0.0017 | 0.0019 | 0.0003 | 0.0008 | 0.0000 | 0.0038 | 0.0011 | 0.8160 |
| Animal husbandry | 0.1040 | 0.0872 | 0.5462 | 0.0288 | 0.0065 | 0.0249 | 0.0004 | 0.1063 | 0.0189 | 0.9233 |
| Fishery | 0.0002 | 0.0007 | 0.0002 | 0.1239 | 0.0001 | 0.0001 | 0.0000 | 0.0007 | 0.0003 | 0.1262 |
| Mining and separating | 0.0003 | 0.0007 | 0.0002 | 0.0001 | 0.0032 | 0.0002 | 0.0000 | 0.0014 | 0.0002 | 0.0062 |
| Manufacturing | 0.0010 | 0.0027 | 0.0008 | 0.0004 | 0.0002 | 0.0019 | 0.0000 | 0.0037 | 0.0006 | 0.0114 |
| Electricity | 0.0008 | 0.0021 | 0.0006 | 0.0003 | 0.0004 | 0.0005 | 0.0043 | 0.0034 | 0.0006 | 0.0130 |
| Construction | 0.0008 | 0.0022 | 0.0006 | 0.0003 | 0.0002 | 0.0004 | 0.0000 | 0.0087 | 0.0009 | 0.0142 |
| Services | 0.0005 | 0.0018 | 0.0004 | 0.0002 | 0.0002 | 0.0003 | 0.0000 | 0.0018 | 0.0015 | 0.0067 |
| **Backward linkages** | 2.7793 | 2.0009 | 1.5501 | 0.6475 | 0.0866 | 0.3151 | 0.0098 | 1.3689 | 0.2418 | 9.0000 |

**Table A3.** Normalized matrix of backward and forward reallocation linkages for green water in economic sectors.

| Sectors | Planting Sector | Forestry | Animal Husbandry | Fishery | Mining and Separating | Manufacturing | Electricity | Construction | Services | Forward Linkages |
|---|---|---|---|---|---|---|---|---|---|---|
| Planting sector | 3.3919 | 1.3971 | 1.2696 | 0.6246 | 0.0959 | 0.3636 | 0.0064 | 1.5743 | 0.2767 | 9.0000 |
| Forestry | 0.0000 | 0.0000 | 0.0000 | 0.0000 | 0.0000 | 0.0000 | 0.0000 | 0.0000 | 0.0000 | 0.0000 |
| Animal husbandry | 0.0000 | 0.0000 | 0.0000 | 0.0000 | 0.0000 | 0.0000 | 0.0000 | 0.0000 | 0.0000 | 0.0000 |
| Fishery | 0.0000 | 0.0000 | 0.0000 | 0.0000 | 0.0000 | 0.0000 | 0.0000 | 0.0000 | 0.0000 | 0.0000 |
| Mining and separating | 0.0000 | 0.0000 | 0.0000 | 0.0000 | 0.0000 | 0.0000 | 0.0000 | 0.0000 | 0.0000 | 0.0000 |
| Manufacturing | 0.0000 | 0.0000 | 0.0000 | 0.0000 | 0.0000 | 0.0000 | 0.0000 | 0.0000 | 0.0000 | 0.0000 |
| Electricity | 0.0000 | 0.0000 | 0.0000 | 0.0000 | 0.0000 | 0.0000 | 0.0000 | 0.0000 | 0.0000 | 0.0000 |
| Construction | 0.0000 | 0.0000 | 0.0000 | 0.0000 | 0.0000 | 0.0000 | 0.0000 | 0.0000 | 0.0000 | 0.0000 |
| Services | 0.0000 | 0.0000 | 0.0000 | 0.0000 | 0.0000 | 0.0000 | 0.0000 | 0.0000 | 0.0000 | 0.0000 |
| **Backward linkages** | 3.3919 | 1.3971 | 1.2696 | 0.6246 | 0.0959 | 0.3636 | 0.0064 | 1.5743 | 0.2767 | 9.0000 |

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
