# Peer review of "Assessment of Inter-Sectoral Virtual Water Reallocation and Linkages in the Northern Tianshan Mountains, China"

_water, doi:10.3390/w12092363_

Round 1

Reviewer 1 Report

Review Request: Round 1 Manuscript ID: water-890034. Title: Assessment of Inter-Sectoral Virtual Water Reallocation and Linkages in the Northern Tianshan Mounntains, China.   This paper uses and input-output method to analyse the reallocation and linkages of virtual water in the economy of the northern Tianshan Mountains, an specific inland arid region in China.   General Comments:   I would like to congratulate the authors for this magnificent and interesting work. The manuscript is well structured and balanced in all its sections. The contribution of this paper to the literature is clear, the geographical area under study is well described, as well as the methodology used. The authors not only describe the results achieved with sufficient tables and figures, but also recommend guidelines to policy makers on how to put the results into practice.

I suggest accepting this paper after considering some minor issues, as detailed below.

Specific Comments:

P. 2, L. 54-57: check this sentence. Is "which" in this sentence necessary?

P. 2, L. 64-69: check the following references to follow the same format throughout the paper (Zhao et al., 2016; Zhang et al., 2019; Yu et al., 2010).

P. 6, L. 186-199: Check the font format. There are two different formats in this paragraph. Is there a reason behind it?

P. 10, L. 280-281: Check the punctuation in this sentence. From my point of view, remove the dot after "(Table 2)".

P. 11, L. 303-304: Consider including the following paper on the competition on the use of freshwater resources for agriculture and electricity generation in Spain, the most arid country in Europe.

- Sesma-Martín, D. (2020). Cooling Water: A Source of Conflict in Spain, 1970–1980. Sustainability, 12(11), 4650.

Reviewer 2 Report

The Authors propose accounting framework for the reallocation of blue and green virtual water (VW) in the economic sector. The approach is described appropriately however the manuscript does not present data transparently. In chapter 2.3 the Authors describe the sources of data but do not provide any. The ‘Instructions for Authors’ clearly state that research data must be openly available. Therefore I suggest to provide relevant data as "Supplementary Files". My concern is also about results of CROPWAT model which are used in the accounting of green water. If the results come from other research it should be cited in the manuscript otherwise the model should be described in the manuscript.

I also advise to revise the manuscript by native English speaker. Some sentences are difficult to understand. Additional specific comments which could improve the manuscript are as follow:

  1. L39 What the Authors mean by ‘assessment objects’.
  2. Please rewrite the sentences in L54-57, L85-88.
  3. L104 Please explain what is ‘RAS method’ and provide reference.
  4. L120 D should be noted as matrix not vector.
  5. L172 The second appearance of ‘backward’ should be replaced with ‘forward’.
  6. L182 Should not the words ‘direct’ be deleted?
  7. L201 The title of paragraph is inappropriate. Please consider this ‘Internal Virtual Water Reallocation’.
  8. L244-246 please rewrite the sentence.
  9. L280 Please delete the full stop at the end of the line.
  10. L280-286 Please avoid repeating in the text the values from the table.
  11. L292 Place the word ‘grain’ in front of ‘crops’. ‘crops such as grain’ makes no sence.
  12. L313 Second ‘consumption’ could be deleted.
  13. L316 ‘realocation out’ to be deleted.
  14. L426 Start with ‘The proposed analysis …’

Reviewer 3 Report

Authors present an interesting work on virtual water allocation very well presented.
The introduction section describes the problem reasonably, the references are relevant and scientifically sound. The methodology is well described, the results are correct and very well presented (I must congratulate the authors for the quality of the graphics) and the discussion and conclussions are adequate.
My concerns mostly relate to methodological questions. In this regard, some unsolved questions should be clarified:
Section 2.2.2: the physical meaning of eq. 1 is not clear. Exactly, why do eq. 1 allow transforming PW in VW? If it were the aim of eq. 1, the authors should make it more clear and explain why such equation allows his transformation.
Section 2.2.3: the authors should clarify why they suppose that effective rainfall coincides with green PW consumtpion. A number or factors could make this assumption unacceptable (deep percolation or the existence of irrigation networks fed from groundwater or natural courses, for example).
Section 2.3: it is not clear how the authors estimate the blue water consumption. Why do they assume that water consumption of the sectors referred to is blue water?
There are some minor typos (see section 2.3).

Reviewer 4 Report

It is a simple but nice piece of research. It uses a standard methodology for a well-known topic, but it is well presented and has some additional scientific soundness.
